# Spatial Variability of COVID-19 Hospitalization in the Silesian Region, Poland

**DOI:** 10.3390/ijerph19159007

**Published:** 2022-07-25

**Authors:** Małgorzata Kowalska, Ewa Niewiadomska

**Affiliations:** 1Department of Epidemiology, Faculty of Medical Sciences in Katowice, Medical University of Silesia, 40-752 Katowice, Poland; mkowalska@sum.edu.pl; 2Department of Epidemiology and Biostatistics, Faculty of Public Health in Bytom, Medical University of Silesia, 41-902 Katowice, Poland

**Keywords:** COVID-19 morbidity, in-hospital mortality, health maps

## Abstract

Assessment of regional variation in the COVID-19 epidemic is an important task for the implementation of effective action in public health, especially in densely populated regions. In this descriptive study, the temporal and spatial variability of morbidity and in-hospital mortality of COVID-19 in the Silesian Voivodship (Poland) was analyzed. Secondary epidemiological data of hospitalized patients due to COVID-19 from 1 March to 31 December 2020 and from 1 January to 31 December 2021 were obtained from the regional registry of the Silesian Voivodship Office in Katowice. A year by year (2020 versus 2021) comparative analysis showed a similar course pattern of the COVID-19 pandemic in the Silesian Voivodeship; with the worst situation occurring in the colder months of the year. The percentage of in-hospital mortality remained at a high level, close to 20% during the second year of observation. The risk of death in patients hospitalized due to COVID-19 increased with the number of comorbidities. The highest number of patients was documented in densely inhabited regions with intensive population movement (Częstochowa and border counties). The epidemiological ‘map’ facilitates the generation of hypotheses needed for the explanation of the observed epidemic hazard in one of the most populated regions of Poland.

## 1. Introduction

Assessment of regional variation in the COVID-19 pandemic is an important task for proper and effective action in public health, e.g., related to the allocation of medical resources. This is especially important in densely populated regions such as Silesian Voivodeship located in the southern part of Poland, where the population density is three times higher than the national average (364 person/km^2^ and 122 person/km^2^, respectively) [1]. The 2021 year was the second year of the COVID-19 pandemic, which began as a serious challenge to existing medical care in many countries including Poland. The severe acute respiratory syndrome coronavirus contributed to significant life expectancy shortening in every country of the world [2]. Current data suggest that life expectancy decreased by 1.6 years in the USA, in most affected European countries: by 1.5 years in Spain, by 1.3 years in Poland, by 1.2 years in Italy, by 1.0 years in the United Kingdom, and 0.6 years in France compared to the previous period 2015–2020 [3]. It is worth noting that the number of excess deaths per million population was particularly high in Poland, as well as in Mexico, the Czech Republic, and Slovakia.

Many prophylactic actions were implemented during the pandemic, including administrative decisions such as changing the functioning of educational institutions and universities to online mode; closure of borders by the State, and the ‘State of the epidemic’ in the whole country was introduced [4]. As a consequence, there were serious restrictions on civil liberties, which were aimed at reducing the number of new infections by maintaining social distancing and observing the sanitary regime. A testing system for SARS-CoV-2 infection was implemented, and contact persons were subject to mandatory quarantine. All these activities were important in the period, in which we had no vaccination and effective treatment of COVID-19 patients. The first vaccinations were introduced in Poland at the end of December 2020, firstly medical staff, employees of social welfare homes and municipal social welfare centers, as well as employees and students of medical universities were vaccinated [5]. One year after the National Vaccination Program was introduced, we can verify the state of vaccination in particular voivodeships and counties. Data from December 2021 reveal a large variation in the percentage of people fully vaccinated against COVID-19 in provinces, from over 70% in counties Mazowieckie, Wielkopolskie, or Zachodniopomorskie to a value below 40% in many counties of the Małopolskie voivodeship. We also observed differences between counties in the Silesian voivodeship, the percentage of fully vaccinated varied from 53% to 66% [6].

Simultaneously, the daily number of hospitalized and dead patients related COVID hospitalization was monitored and recorded in the government registry [7] which enables the tracking of temporal and spatial variability of hospitalized morbidity and in-hospital mortality in COVID-19 patients. A high density of population, low percentage of fully vaccinated inhabitants, and an insufficient number of medical care workers in Polish hospitals (physicians and nurses) [8,9] observed before a pandemic are probably important factors affecting the level of the observed epidemiological situation.

Mentioned arguments lead to a necessity of an in-depth analysis of COVID-19 hospitalization in one of the largest agglomerations in Poland inhabited by a 4.5 mln population. Without a detailed analysis of the temporal and spatial variability of patients’ numbers and their chances of survival, it will not be possible to effectively counteract epidemic crises in the future. In such a situation, it is necessary to refer to well-known, recognized methods of descriptive biostatistics, including visualization methods (maps) that are needed to help direct health policy, resource allocation, research, and patient care [10,11].

## 2. Materials and Methods

In this descriptive type of study, the temporal and regional variability of morbidity and in-hospital mortality of COVID-19 in the Silesian Voivodship was presented. Secondary epidemiological data including sex, age, place of residence in particular counties (poviats), comorbidities, and discharge mode of hospitalized patients were obtained from the registry of the Silesian Voivodship Office in Katowice. The analyses were conducted separately for two periods: from 1 March to 31 December 2020 (previously published) and between the 1st of January to the 31st of December 2021. Such a proceeding allows comparing both years of the COVID-19 pandemic in the Silesia region, before and during the campaign for SARS-CoV-2 vaccination. The hospitalized morbidity was presented by a crude coefficient calculated as the cumulative number of hospitalizations per 100,000 population [12]. The hospitalized morbidity rates were standardized using the European population as a reference [13]. In-hospital mortality was calculated as the percentage of death occurring in patients during the hospital stay. Morbidity and in-hospital mortality were presented for both 2020 and 2021. Development and the availability of spatial analytical techniques, based on geographic information systems (GIS), enable the visualization of epidemiological data. This was important in the case of infectious diseases because such created maps allow for the recognition of the spatial severity of health problems and quick reaction in public health [14]. The study wasn’t any medical experiment, and the secondary character of data didn’t need Bioethics Committee permission.

The results were presented as an arithmetic mean and standard deviation (X ± SD) for quantitative variables and as numbers and percentages (*n*, %) for qualitative variables. The distribution of variables was assessed using the Shapiro-Wilk test. The Mann-Whitney U-test was used to assess the differences between quantitative variables and the Chi-square test for qualitative variables, adequately. The correlation coefficient R’ Spearman was used to assess relationships between social or demographic factors and the number of hospitalized patients due to COVID-19. The analysis used the capabilities of the Statistica 13.3 software, Stat Soft Poland (Cracovia, Poland) (descriptive data and difference tests), and the access to the QGIS 3.16 geographic information systems, Open Source Geospatial (OSGeo) (for the presentation of spatial variability). Multivariate spatial analysis was developed for counties of Silesian Voivodeship with use of generalized linear regression (GLR). The Moran’s I index with contiguity edges corners conceptualization, and its significant test were used to present spatial autocorrelation between counties of Silesian Voivodeship for standardized rate of hospitalized morbidity, separately in 2020 and 2021. Calculations were made using geoprocessing tools of ArcGIS Pro v.2.9.0, ESRI, 2021 (Warsaw, Poland). Statistical significance was based on the criterion *p* < 0.05.

## 3. Results

The study results suggest that the number of COVID-19 hospitalized and dead patients in the Silesian voivodeship varied depending on the season, the highest level remained during colder months of both compared years (Table 1). Moreover, the highest value of in-hospital mortality (exceeding 20%) was noted in the spring (March-April) and autumn (October-December) of the 2020 year. In the second year of the COVID-19 pandemic, monthly differences were smaller while in-hospital mortality varied from 18% in June and 24.3% in April or 24.0% in December (Figure 1).

Moreover, the risk of death in patients hospitalized due to COVID-19 increased with the number of comorbidities (Table 2). The highest odds ratio was obtained for patients with three coexisting diseases within the group of comorbidities: respiratory diseases (36.7%), cardiovascular (32.1%), metabolic (11.3%) diseases, and diseases of the genitourinary system (4.8%).

Finally, the spatial variability of the number of hospitalized and dead patients due to COVID-19 was also analyzed. The results indicated that in both years the highest number of COVID-19 hospitalized patients was noted in Cieszynski county, as well as in the large cities of Czestochowa, Katowice, Sosnowiec, Gliwice, Bielsko-Biała, and Rybnik (Appendix A). A significantly higher number of hospitalized COVID-19 patients was observed in densely inhabited municipalities; with a Spearman’s correlation coefficient of 0.68 (*p* < 0.0001) and 0.75 (*p* < 0.0001) in 2020 and 2021, respectively (Appendix A). Additionally, we confirmed that the number of patients over 65 years of age was significantly higher than those younger (except for the Rybnicki region) during the study period (from March 2020 to December 2021). Detailed data were presented in Appendix A. A weak and not statistically significant correlation between the number of patients hospitalized due to COVID-19 and social or demographic factors in particular counties of the study region was confirmed by GLR model (Table 3).

The value of cumulatively hospitalized morbidity in Silesian Voivodeship in the 2020 year was at the level of 366.7 per 100,000 inhabitants, while in 2021 it was twice as high (779.7 per 100,000). In both years, the highest crude rates of morbidity were documented in Cieszynski county (1124.3/100,000 and 987.4/100,000 inhabitants, in 2020 and 2021, respectively), and the city of Rybnik (1099.7/100,000 and 862.5/100,000 inhabitants, respectively). Detailed data are presented in Appendix A. The standardized hospitalized morbidity rates in 2020 and 2021 were at the level of 250.6/100,000 and 537.0/100,000 inhabitants, respectively. Similarly, the highest values were recorded in Cieszynski county (745.5 and 704.9/100,000, respectively), Rybnicki county (685.6 and 639.3/100,000), and the city of Rybnik (878.2 and 630.3/100,000) in both years (Figure 2). Significant, positive Moran’s index value at the level I = 0.32 (*p* < 0.001) for the standardized rate of hospitalized morbidity in the 2020 year indicates a tendency toward clustering, while in 2021 the spatial autocorrelation was at the level I = 0.16 (*p* = 0.06).

When assessing the case fatality rate among COVID-19 patients, the highest values in 2021 (similar to 2020) was evident in those hospitalized in larger cities such as Czestochowa, Katowice, Gliwice, and Cieszynski county (Appendix A). As we previously mentioned, comorbidities significantly increased the risk of in-hospital mortality (Table 2). The percentage of in-hospital mortality in COVID-19 patients in the Silesian Voivodeship was higher in 2021 (24.8%) than in 2020 (17.4%). Again, the value of in-hospital mortality varied in the studied subregion, the highest percentage was recorded in Jastrzębie-Zdrój (32.7%), Jaworzno (32.6%), Tychy (30.0%), and the Raciborski county (29.6%). Additionally, we confirmed that the majority (32 from 36 counties) of in-hospital mortality exceeded 20%. Simultaneously, we noted a lower fatality case ratio in 2021 in many counties in which the number of hospitalized was high from the beginning of COVID-19 pandemic (Czestochowa, Katowice, Sosnowiec, Bielsko-Biala, czestochowski, cieszynski, and rybnicki county) compared to other subregions of Silesian voivodeship. The correlation coefficients for the relationship between in-hospital mortality due to COVID-19 in 2021 and the number of medical staff (physicians and nurses per 10,000 inhabitants) were low and not statistically significant. They remain at the level R’ = 0.26 (*p* = 0,1) and R’ = 0.17 (*p* = 0.3), respectively, for physicians and nurses.

## 4. Discussion

The results from the study confirmed that secondary epidemiological data are useful in the recognition of the course of the epidemiological situation of infectious diseases such as COVID-19 in one of the most densely populated regions of Poland. Analysis of registry data related to daily hospitalization and in-hospital mortality in 2020 and 2021 allowed us to assess the temporal and spatial variability of the epidemic in Silesian voivodeship, and recognize how the course of the epidemic changed after the introduction of preventive vaccinations in the same region. This diagnosis is very important to future public health preventive actions due to a very easy way of transmitting infections in the population and a simultaneous increase in the number of COVID-19 hospitalized patients in the colder season of the year [12]. There is an expectation that the global effort in vaccination and effective surveillance with an adequate population response could protect against the next epidemic [15]. 

The authors of the last cited publication indicate that the most likely scenario will be the transition to an epidemic seasonal disease such as influenza. The results from our analysis confirmed that the course of the COVID-19 epidemic in both compared years in the Silesian voivodeship was similar to that of seasonal influenza with a maximum of infections in colder months of the year [16]. The previous publications from our team suggest that vaccination against seasonal influenza was negatively associated with SARS-CoV-2 infection in the studied region [17]. Numerical simulation of available data from ten countries of the world highlighted the necessity of considering seasonal factors when formulating future intervention strategies [18]. Analysis of Polish data is oriented at the number of hospitalized patients and the risk of in-hospital mortality in the Silesian voivodeship before and after the National Vaccination Program against COVID-19. It is worth underlining that the first case of COVID-19 in Poland was confirmed in March 2020 and from that moment we noted a slow and gradual implementation of diagnostic tests which had an impact on the increase in the number of registered cases. From June to November 2021 we noted a significant reduction in both values (hospitalization and in-hospital mortality). However, the number of yearly cases was higher in comparison to the first year of the epidemic. The number of hospitalized cases and deaths was two times higher in December 2021 than in December of the previous year (2020). Additionally, in the second year of observation, the percent of in-hospital mortality remained at a high level, close to 20% which was typical for the colder months of the previous year. It was confirmed that the in-hospital mortality rate observed in Silesian voivodeship was higher than the values reported by other authors [19,20]. It is worth mentioning that the development of population immunity is a long-term process. At the beginning of the 2021 year, vaccination was available only for medical and social workers and the percentage of fully vaccinated people at the end of the year was not satisfactory in Poland. Moreover, official data confirmed a serious variability in the frequency of fully vaccinated with the highest percentage noted in Mazowieckie, Wielkopolskie, or Zachodniopomorskie voivodeship and the lowest value in Małopolskie voivodeship. We observed similar differences in counties of the Silesian voivodeship, with the percentage of fully vaccinated individuals varying from 53% to 66% [6]. Most likely, this situation may result in an adverse differentiation of hospitalized morbidity and mortality in the counties of the studied region. It was also documented that either one dose of Pfizer-BioNTech or AstraZeneca vaccines was associated with a significant reduction in symptomatic COVID-19 in older adults and subsequent protection against severe disease [21].

On the other hand, a high and persistent in-hospital mortality rate may be an effect of an inefficient medical care system, especially related to significant shortages of medical staff in hospitals [22,23] in the Silesian voivodeship. The main determinants of physicians’ and nurses’ retention include inter alia disproportionately lower earnings, the sinister atmosphere at work, and a significantly increased workload due to lack of medical staff, and bureaucracy [24]. Further assessment reveals that the migration of the young generation of nurses and doctors from Poland should become a key element of the human resources policy in the Polish health care system [25]. Furthermore, It was documented that a high hospital load in Israel significantly increased COVID-19-related mortality compared to periods of lower patient load [26]. Other authors speculated that, due to the public health restrictive strategy, people with minor health problems avoided hospital visits as a result, and patients with the most serious health problems were hospitalized and the risk of death in this group was the highest [27,28]. On the other hand, we observe the high variability of the coronavirus, the Delta plus mutation has become dominant in the EU countries, including Poland. Unfortunately, this type of SARS-CoV-2 virus is more virulent and leads to more serious health consequences, including death [29].

The results obtained from our study confirmed a statistically significant relationship between the number of coexisting diseases and the risk of death in hospitalized due to COVID-19 in the Silesian voivodeship. The presence of three comorbidities increased the odds ratio in hospitalized patients by more than 4 times in 2020 and more than 2.5 times in 2021. The most frequently recorded coexisting diseases were respiratory diseases (J00-J99) and cardiovascular diseases (I00-I99 according to ICD-10). Obtained results are in line with the opinion that patients with diabetes, chronic obstructive pulmonary disease, cardiovascular diseases, hypertension, malignancies, and other comorbidities could develop a life-threatening situation [30]. Current published data confirmed that COVID-19 plays an exacerbating role in the worsening of health status in patients with pre-existing liver diseases [31]. Moreover, the mean value of the age of deceased patients in Silesian hospitals was close to 75 years which was comparably similar to those observed in other studies [32,33,34,35,36]. Regardless of the study region, it was shown that the presence of comorbidities in older patients was a factor leading to a more severe course of the disease and thus increased the risk of death [37,38,39]. On the other hand, the observed decreasing mortality ratio in COVID-19 patients hospitalized in New York from March to August 2020 could be an effect of the increasingly younger age of patients with fewer comorbidities [28]. Results of other observations suggest that the COVID-19 scoring based on simple demographic (sex, age) and laboratory parameters (e.g., hemoglobin, platelets, leukocytes, creatinine, C-reactive protein CRP), and the occurrence of chronic diseases may become a widely accessible and objective tool for predicting mortality in hospitalized patients [40,41,42]. However, the spatial variability of hospitalized morbidity in the Silesian voivodeship was observed even after ratio standardization.

## 5. Conclusions

We can conclude that the secondary epidemiological data based on the MZ/Szp-11 form are an interesting source allowing for the diagnosis of the COVID-19 epidemiological situation in one of the most densely populated regions of Poland. They present the dynamics of health services in hospitalized patients in particular months of the year. At the same time, they make it possible to track the spatial variability between counties of the Silesian voivodeship both morbidity and in-hospital mortality. They also allow recognizing determinants related to sex, age of patients, and coexisting diseases to predict the potential output of treatment. The presented epidemiological ‘map’ facilitates the generation of hypotheses needed for the explanation of the observed spatial and seasonal variability of the risk. In our opinion, mapping is a very useful source of data in public health which can help recognize actual health needs related to the new epidemiological hazard, i.e., COVID-19.

## Figures and Tables

**Figure 1 ijerph-19-09007-f001:**
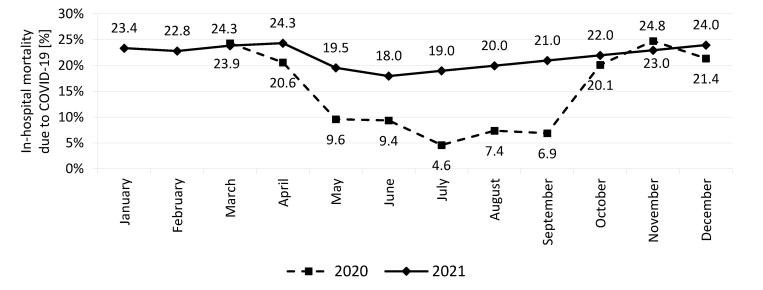
In-hospital mortality due to COVID-19 in the Silesian voivodeship in particular months of both years (2020 and 2021).

**Figure 2 ijerph-19-09007-f002:**
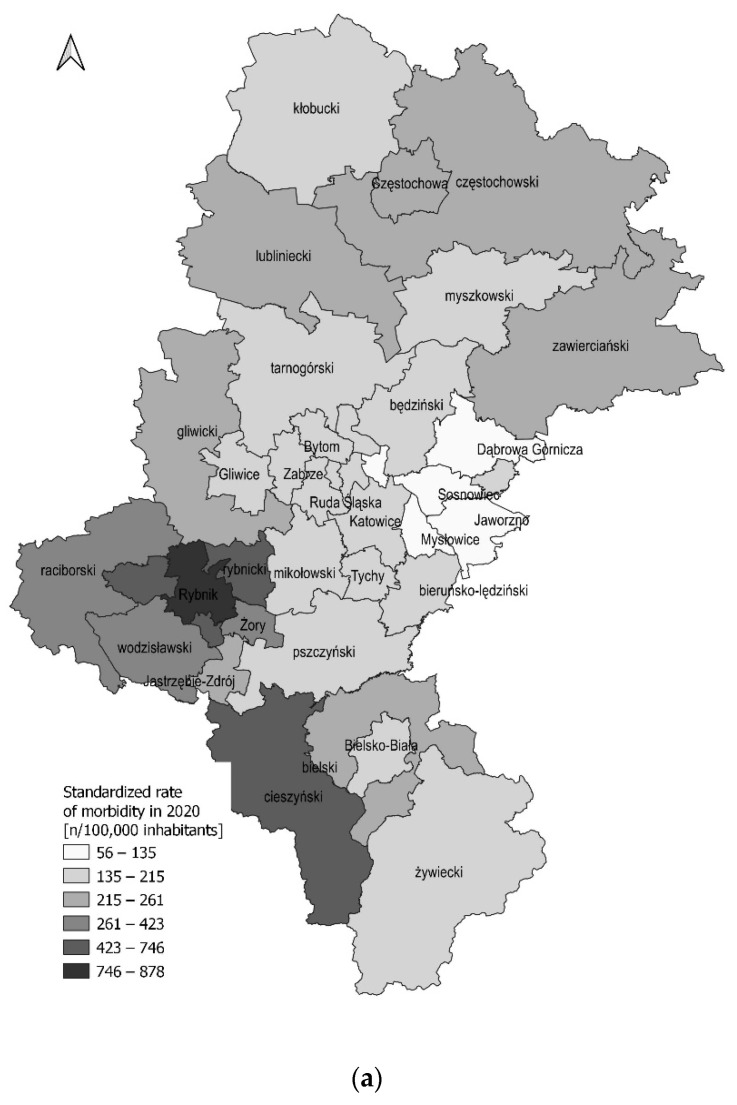
Standardized rate of hospitalized morbidity (n/100,000) due to COVID-19 in hospitals of the Silesian Voivodeship by counties in the years (**a**) 2020 and (**b**) 2021.

**Table 1 ijerph-19-09007-t001:** Number and the average age of monthly hospitalized and dead COVID-19 patients in the Silesian voivodeship in the years 2020–2021 by the particular months.

Month	The Year 2020					
	Women *n* (%)	Men *n* (%)	Hospitalized Patients *n* (%)	Dead Patients *n* (%)	Age of Hospitalized Patients X ± SD	Age of Dead Patients X ± SD
January	-	-	-	-	-	-
February	-	-	-	-	-	-
March	15 (0.2)	22 (0.2)	37 (0.2)	9 (0.3)	57.5 ± 20.5	72.9 ± 9.9
April	227 (3)	234 (2.5)	461 (2.7)	95 (3.2)	62.4 ± 17.9	76.4 ± 11.6
May	516 (6.9)	516 (5.4)	1032 (6.1)	99 (3.3)	58.4 ± 19.3	73.8 ± 11.6
June	613 (8.2)	689 (7.2)	1302 (7.7)	122 (4.1)	57.1 ± 19.8	72.8 ± 12.9
July	548 (7.3)	666 (7)	1214 (7.1)	56 (1.9)	59.2 ± 20.6	74.4 ± 12.7
August	549 (7.3)	683 (7.2)	1232 (7.2)	91 (3.1)	59 ± 20.8	75.7 ± 11.1
September	415 (5.5)	529 (5.6)	944 (5.6)	65 (2.2)	61.9 ± 20.6	73.2 ± 13.2
October	1000 (13.3)	1225 (12.9)	2225 (13.1)	448 (15.1)	63.9 ± 19	75.9 ± 10.7
November	1799 (24)	2758 (29)	4557 (26.8)	1128 (38)	67.1 ± 15.9	73.5 ± 11.6
December	1810 (24.2)	2189 (23)	3999 (23.5)	855 (28.8)	67.1 ± 15.7	74.4 ± 11.3
Total	7492 (100)	9511 (100)	17,003 (100)	2968 (100)	63.8 ± 18.3	74.3 ± 11.5
**Month**	**The Year 2021**					
	**Women** * **n** * **(%)**	**Men** * **n** * **(%)**	**Hospitalized** **Patients** * **n** * **(%)**	**Dead** **Patients** * **n** * **(%)**	**Age of** **Hospitalized** **Patients** **X ± SD**	**Age of** **Dead** **Patients** **X ± SD**
January	1420 (8.6)	1661 (8.6)	3081 (8.6)	720 (8.1)	68.8 ± 14.9	75.7 ± 11.3
February	1028 (6.2)	1318 (6.8)	2346 (6.6)	535 (6)	68.4 ± 15.3	75.5 ± 10.8
March	2611 (15.8)	3117 (16.2)	5728 (16)	1366 (15.4)	65.9 ± 15.7	73.7 ± 10.8
April	4266 (25.8)	5468 (28.4)	9734 (27.2)	2369 (26.7)	66 ± 14.9	72.9 ± 11.4
May	1876 (11.4)	2213 (11.5)	4089 (11.4)	799 (9)	67.2 ± 15.1	73.8 ± 12.1
June	323 (2)	306 (1.6)	629 (1.8)	113 (1.3)	66.6 ± 16.3	73.8 ± 11.7
July	68 (0.4)	52 (0.3)	120 (0.3)	14 (0.2)	64.2 ± 18.8	74.5 ± 13.9
August	48 (0.3)	49 (0.3)	97 (0.3)	16 (0.2)	65.1 ± 19.5	79.6 ± 8.1
September	80 (0.5)	76 (0.4)	156 (0.4)	23 (0.3)	64.9 ± 18.9	79 ± 11.8
October	269 (1.6)	272 (1.4)	541 (1.5)	138 (1.6)	64.3 ± 19.8	75.7 ± 12
November	1334 (8.1)	1407 (7.3)	2741 (7.7)	799 (9)	63.4 ± 21.6	75.3 ± 11.7
December	3205 (19.4)	3304 (17.2)	6509 (18.2)	1967 (22.2)	65.9 ± 18.8	74.4 ± 11.9
Total	16,528 (100)	19,243 (100)	35,771 (100)	8859 (100)	66.3 ± 16.6	74.1 ± 11.5

Legend: *n*—number, X—mean value, SD—standard deviation, - —no data in January and February 2020.

**Table 2 ijerph-19-09007-t002:** The odds ratio explained the risk of death in COVID-19 patients according to the number of coexisting diseases.

The Odds Ratio of Death in COVID-19 Hospitalized Patients According to the Number of Coexisting Diseases				
Reference Group: Patients without Comorbidities	The Year 2020		The Year 2021	
	Odds Ratio	95% Confidence Interval	Odds Ratio	95% Confidence Interval
One disease	1.10	0.98–1.24	1.16	1.08–1.24
Two diseases	3.14	2.79–3.53	1.97	1.83–2.12
Three diseases	4.57	4.11–5.08	2.57	2.41–2.74

**Table 3 ijerph-19-09007-t003:** Correlation between the number of patients hospitalized due to COVID-19 in hospitals of the Silesian Voivodeship in 2020 and 2021, and social or demographic factors by counties.

Social or Demographic Factors	2020			2021		
	R’ *p*-Value	GLR Model		R’ *p*-Value	GLR Model	
		B *p*-Value	VIF		B *p*-Value	VIF
Average age of patients hospitalized due to COVID-19	0.05 *p* = 0.76	−34.0 *p* = 0.22	1.62	0.45 *p* = 0.006	100.97 *p* = 0.06	1.09
Average number of coexisting diseases	−0.47 *p* = 0.004	−367.61 *p* = 0.11	1.37	−0.24 *p* = 0.15	−201.61 *p* = 0.50	1.07
Number of physicians per 10,000 inhabitants	−0.01 *p* = 0.95	3.35 *p* = 0.55	12.54	0.31 *p* = 0.06	17.87 *p* = 0.08	10.52
Number of nurses per 10,000 inhabitants	0.09 *p* = 0.61	0.83 *p* = 0.17	10.26	0.26 *p* = 0.11	−6.21 *p* = 0.22	9.06
Percentage of urban residents	−0.18 *p* = 0.29	−3.59 *p* = 0.27	2.64	0.09 *p* = 0.6	−5.32 *p* = 0.14	2.45
Population density per 1 km^2^	−0.1 *p* = 0.56	−0.03 *p* = 0.75	2.28	0.12 *p* = 0.48	−0.09 *p* = 0.39	2.15
Diagnosis GLR	-	R^2^ = 0.14 AIC = 538.6		-	R^2^ = 0.36 AIC = 547.5	

Legend: R’—Spearman coefficient; GLR—Generalized Linear Regression; B—linear coefficient; VIF—Variance Inflation Factor (VIF); R^2^—coefficient of determination; AIC—Akaike information criterion.

## Data Availability

Official registry of the Silesian Voivodship Office in Katowice, Poland.

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
