# Peer review of "Spatial Variability of COVID-19 Hospitalization in the Silesian Region, Poland"

_ijerph, 2022, doi:10.3390/ijerph19159007_

Round 1

Reviewer 1 Report

However, I consider this manuscript to have valuable data that would be of interest if published, it needs a major revision before further consideration of publication process. The major concerns are: 1. The statistical analysis is too simple i.e. first the authors should use Moran Index to analyze the spatial dependence between the counties of the COVID-19 infections. 2. There is a lack of any analysis of social or demographic factors and the number of COVID-19 infections in the counties like mean age, percentage of urban residents, number of GPs or hospital beds, co-morbidities (the SAR model should be used) 3. Did the authors obtaine the ethical approval for the present study.

Author Response

First, we would like to thank for this very detailed analysis of the manuscript. We agree with the majority of the Reviewers’ comments. Our replies are written in a red. We made several changes in our manuscript believing they address all the Reviewers’ comments (also marked in red). Please see the attachment.

Reviewer 2 Report

Dear authors,

This article highlights the impact of the pandemic Covid-19 infection in Poland, in different regions of the country.

The results are presented in detail in different regions of the country, over the two years and correlated with the age of the patients and with different comorbidities, which brings a complete picture of the pandemic evolution in Poland.

The bibliography is recent and relevant for this study and brings an important addition to this article.

I suggest some aspects that can be improved:

Lines 35-36: Acute respiratory syndrome, due to coronavirus infection, significantly reduces life expectancy. I suggest to the authors to highlight the impact on some European countries that have been significantly affected by this pandemic, because apart from the USA, the only country in Europe was Spain (except for Poland of course, which is the subject of this study).

Lines 244-246: I suggest the authors develop this paragraph which states that the statistical evaluation of demographic and laboratory parameters can become an accessible tool for predicting mortality in inpatients. In the context in which the afferent bibliographic title refers to SARS-VOC-2 infection, possibly supplementing with citation other studies to support this claim.

Author Response

(The authors gave the same response as above.)

Reviewer 3 Report

Lessons from the management of the COVID-19 pandemic can be properly learned if epidemiological data is widely processed and analyzed. It is very important to get to know the geographical and social factors that influence mortality, and within this to explore regional or national variations.

This descriptive study confirms the correlation of population density and certain comorbidities with the rate of COVID-19 mortality in a Polish Voivodeship. The methods used are appropriate, the draft is well-understood and clearly worded.

The discussion raises several circumstances that require further research, such as the staffing situation of the health services. 

Author Response

We would like to thank you for your kind opinion.

Round 2

Reviewer 1 Report

I am satisfied with authors' reply